# An Electro-Mechanical Actuator Motor Voltage Estimation Method with a Feature-Aided Kalman Filter [note 1]

**DOI:** 10.3390/s18124190

**Published:** 2018-11-29

**Authors:** Yujie Zhang, Liansheng Liu, Yu Peng, Datong Liu

**Affiliations:** Department of Automatic Test and Control, Harbin Institute of Technology, Harbin 150001, China; hnhyzyjlh@hit.edu.cn (Y.Z.); lianshengliu@hit.edu.cn (L.L.)

**Keywords:** electro-mechanical actuator, performance degradation, voltage estimation, feature-aided Kalman filter

## Abstract

Electro-Mechanical Actuators (EMA) have attracted growing attention with their increasing incorporation in More Electric Aircraft. The performance degradation assessment of EMA needs to be studied, in which EMA motor voltage is an essential parameter, to ensure its reliability and safety of EMA. However, deviation exists between motor voltage monitoring data and real motor voltage due to electromagnetic interference. To reduce the deviation, EMA motor voltage estimation generally requires an accurate voltage state equation which is difficult to obtain due to the complexity of EMA. To address this problem, a Feature-aided Kalman Filter (FAKF) method is proposed, in which the state equation is substituted by a physical model of current and voltage. Consequently, voltage state data can be obtained through current monitoring data and a current–voltage model. Furthermore, voltage estimation can be implemented by utilizing voltage state data and voltage monitoring data. To validate the effectiveness of the FAKF-based estimation method, experiments have been conducted based on the published data set from NASA’s Flyable Electro-Mechanical Actuator (FLEA) test stand. The experiment results demonstrate that the proposed method has good performance in EMA motor voltage estimation.

## 1. Introduction

In recent years, compared with traditional hydraulic technologies, more fly-by-wire ones are utilized in aerospace applications [1,2,3,4,5]. Fly-by-wire flight control actuation is lighter and more reliable. Importantly, its maintenance operation is easier to be conducted than hydraulic flight control actuation. Electro-Mechanical Actuator (EMA) is a typical fly-by-wire flight control actuation [6]. Small EMAs have been incorporated in some aircrafts for their secondary functions, such as trim tab actuation and spoiler [7]. Efforts have been made to deploy EMA for utility roles, such as landing gear, weapon bay door, and aerial refueling door, in F-35 Joint Strike Fighter, Airbus 380 and Boeing 787. The architecture of future aircraft will incorporate more energy-efficient EMAs for their flight control actuation [8]. However, there is a high demand for EMA safety and reliability as an actuator is one of the most safety-critical components in aircraft. Undetected actuator faults can lead to catastrophic consequences of aircraft. For example, Alaska Airlines MD-83 Flight 261crashed due to the failure of the horizontal stabilizer EMA due to excessive wear and insufficient lubrication of its jack screw [9].

In fact, EMA components are relatively new in aerospace applications and thus have not been deployed in sufficient amounts and time to accumulate reliable fault statistics. With EMA increasingly utilized in safety-critical applications, the development of reliable and accurate EMA prognostic health management (PHM) systems becomes more and more important [8,10,11,12]. Development of EMA PHM can also contribute to increasing the overall availability and safety of EMA, in which performance degradation, faults and catastrophic failures can be prevented [13,14,15]. Furthermore, EMA PHM also makes EMA more suitable for aerospace applications due to the ability of minimizing failures [16,17,18].

EMA PHM approaches can be classified into two major categories: model-based approaches and data-driven approaches [19,20,21]. Generally, model-based approaches require an accurate mathematical model to predict the outputs related to some inputs for health assessment. For instance, the deviation between the outputs of the accurate mathematical model and the actual outputs can be utilized to estimated system parameters, such as efficiency and damping. Through comparing the estimated parameters and the parameters of a healthy system, the EMA fault or degradation state can be identified [10,22]. There are also some works that have extensively studied parameter estimation utilizing extended Kalman Filter and observer, in which complex algorithms are developed for high-accuracy parameter estimation [23,24,25]. The advantage of model-based approaches is that there is a clearly correspondence between failure modes and model parameters. However, there are also some challenges of model-based approaches. One of the most common challenges is that models utilized in model-based approaches are often too complex to be obtained. As a result, for model-based approaches, the models are required to be very specific and each new application must have a validated model.

In contrast, data-driven approaches can provide insight into machine condition through applying signal processing techniques directly into the signal data [26]. Data-driven approaches have no requirement for complex models and can be utilized in many systems with different types. Therefore, it becomes an attractive option in EMA PHM. For instance, Balaban et al. propose a variety of condition indicators that enabled detection, identification, and isolation among various fault modes based on monitoring data [6]. These indicators include Temperature Deviation (TD), Drift Indicator (DI), Signal Standard Deviation (SD), Load Profile Indicator and Force Indicator. Bodden et al. utilize vibration sensors to indicate fault frequencies which will change with components wearing [27]. The overlap degree of signal probability densities between “aged” EMAs and the “healthy” EMA can also be utilized to assess EMA condition [28]. Additionally, there also exist other studies that are related to health classification and prognostics. For example, Byington et al. apply Fast Fourier Transformation (FFT)-based feature techniques into neural network-based error tracking methods [29]. Then, fuzzy logic and Kalman filter are utilized in their health classification and prognostics model, respectively. Brown et al. propose another health prognostics approach in which the Hilbert transformation is utilized for identifying EMA motor turn-to-turn winding faults and the particle filter is introduced for EMA anomaly detection and prognostics [30]. The remaining useful life prediction of EMA utilizing data-driven methods is also studied [31]. Beyond that, many studies also focus on on motor voltage-based EMA PHM. For instance, Qiao et al. point out that the relationship between actuation load and motor voltage is required to construct an accurate EMA model [32]. Balaban et al. propose that voltage sensor output can be utilized in EMA observer-based diagnosis [33]. Smith et al. point out that the effective number of motor windings utilized for generating the health indicator can be extracted from motor voltage monitoring data [10]. Motor voltage monitoring data can also be affected by EMA motor winding inductance reduction [34]. These studies reveal that developing accurate EMA motor voltage estimation approaches is critical for EMA PHM. Currently, many studies related to this field are also publicly reported. Srivastava et al. introduce a neural networks-based method in power systems’ voltage estimation [35]. Noguchi et al. implement the power-source voltage estimation through a Pulse-Width Modulation (PWM) converter physical model [36]. A digital filter technique can also be utilized in power systems’ voltage estimation [37]. Yu et al. present a voltage estimation method for PWM inverter based on dead-time compensation [38]. A scheme of voltage estimation is proposed with a full-order estimator for PWM rectifiers [39]. Aguilera et al. propose a discrete Kalman Filter (KF) for the capacitor voltage estimation of multilevel converters [40]. However, there are still some challenges in the application of data-driven approaches.

One of the main challenges is the problem caused by non-constant and differing operating conditions. For instance, with various motor speed and load, fault frequencies will change which lead to much difficulty for fault identification. Simultaneously, EMA motor voltage estimation under non-constant and differing operating conditions is still a problem. In fact, EMA motor voltage is susceptible to the electromagnetic interference (EMI) caused by operating condition changes [41,42,43]. The common-mode voltage generated by EMI included in the EMA motor voltage monitoring data will lead to deviation between motor voltage monitoring data and real motor voltage. Fortunately, as a popular method, KF shows a good performance in voltage estimation utilizing monitoring data and a state equation. However, the state equation of EMA motor voltage is not available due to the complexity of EMA. As a result, a new method known as the Feature-aided Kalman Filter (FAKF) is proposed to address this problem and obtain a better voltage estimation. FAKF is an improved KF, in which an aided feature is introduced. The physical model of the aided feature and the feature to be estimated is used to substitute the essential KF state equation. In this way, when the KF state equation is not available or difficult to obtain, the improved KF can still be implemented. In this study, a physical model of EMA motor current and voltage is introduced to FAKF. Then, the data of model-based voltage estimation and voltage monitoring data are utilized to obtain a better EMA motor voltage estimation through FAKF. This method is proposed to meet the requirement of accurate estimation of EMA motor voltage.

The rest of this paper is organized as follows. Section 2 introduces the relevant theories for FAKF. Section 3 presents the framework of FAKF-based voltage estimation. Section 4 illustrates the experiment results and analysis. Section 5 concludes this study and presents our future studies.

## 2. Relevant Theories

### 2.1. Kalman Filter

KF is one of the most well-known estimation methods utilizing noisy measurements and a linear model, which is proposed by Rudolph E. Kalman in 1960 [44,45]. The KF is traditionally utilized to estimate the discrete-time process state and represented as transition equation [46],
(1)$$x_{k}=A_{k-1}x_{k-1}+w_{k-1},$$
where $x_{k}\in R^{n}$ and $A_{k-1}$ denote the dynamic state at current time step *k* and the dynamic model matrix at previous time step k−1, respectively. The measurements can be represented as follows:(2)$$y_{k}=H_{k}x_{k}+v_{k},$$
where $y_{k}\in R^{d}$ and $H_{k}$ denote the measurements and the measurement matrix at current time step *k*, respectively.

In Equations (Equation 1) and (Equation 2), $w_{k-1}$ and $v_{k}$ are random variables, which denote the process noise at at previous time step k−1 and measurement noise at current time step *k*, respectively. Generally, process noise *w* and measurement noise *v* are both assumed to obey Gaussian distribution:(3)$$p\left(w\right)∼N\left(0,Q_{k}\right),$$
(4)$$p\left(v\right)∼N\left(0,R_{k}\right),$$
where N(•) represents multivariate Gaussian distribution. $Q_{k}$ and $R_{k}$ matrices represent process noise covariance and measurement noise covariance at current time step *k*, respectively.

Based on the theories mentioned above, KF can also be simplified as follows [47]:(5)$$x_{k}∼N\left(A_{k-1},Q_{k}\right),$$
(6)$$y_{k}∼N\left(H_{k}x_{k},R_{k}\right).$$

In fact, $x_{k}$ is an unknown exact value of current state, while $y_{k}$ is an observed approximate value of it. Furthermore, $A_{k}$, $Q_{k}$, $H_{k}$ and $R_{k}$ are assumed to be known constant parameters. The KF-based estimation method mainly includes the following two steps:

**Step 1:** Prediction step:(7)$$m_{k}^{-}=A_{k-1}m_{k-1},$$
(8)$$P_{k}^{-}=A_{k-1}P_{k-1}^{-}A_{k-1}^{T}+Q_{k-1}.$$

**Step 2:** Update step:(9)$$K_{k}=P_{k-1}^{-}H_{k}^{T}{\left(H_{k}P_{k-1}^{-}H_{k}^{T}+P_{k}\right)}^{-1},$$
(10)$$m_{k}=m_{k}^{-}+K_{k}\left(y_{k}-H_{k}m_{k}^{-}\right),$$
(11)$$P_{k}=P_{k}^{-}-K_{k}\left(H_{k}P_{k-1}^{-}H_{k}^{T}+P_{k}\right)K_{k}^{T}.$$

In Equations (Equation 7)–(Equation 11), $m_{k}^{-}$ represents priori state mean value and $m_{k}$ denotes the posteriori state mean value. $P_{k}^{-}$ and $P_{k}$ denote the priori state covariance and the posteriori state covariance, respectively. $K_{k}$ is named Kalman gain.

From the introduction of KF, it can be concluded that the estimation can be conducted by utilizing KF and monitoring data. However, obtaining the KF transition equation for EMA motor voltage is difficult due to the complexity of EMA. Hence, a new method named FAKF is proposed in this study to obtain the accurate estimation of EMA motor voltage.

### 2.2. Feature-Aided Kalman Filter

In KF, the transition equation is essential. As for the KF-based voltage estimation, the voltage estimation at current time step *k* can be expressed as follows:(12)$$V_{k}=f\left(V_{k}^{-},VM_{k}\right),$$
where $V_{k}$ and $V_{k}^{-}$ denote posteriori voltage estimation and priori voltage estimation at current time step *k*, respectively. $VM_{k}$ is the voltage measurement at current time step *k*. f(•) indicates the function of $V_{k}$ and $V_{k}^{-}$, $VM_{k}$.

However, KF cannot be directly utilized to estimate the EMA motor voltage because constructing the transition equation about EMA motor voltage is difficult. In fact, there is no clear relationship between EMA motor voltage values at current time step *k* and previous time step k−1. In this study, a new method, named FAKF, is proposed to address this problem. In FAKF, a physical model of EMA motor voltage and current is introduced to substitute KF transition equation. Generally, in KF-based voltage estimation, priori voltage estimation $V_{k}^{-}$ is required to obtain posteriori voltage estimation $V_{k}$, which is calculated based on $V_{k-1}^{-}$ and $A_{k-1}^{-}$. As for EMA motor voltage estimation, $A_{k-1}^{-}$ is unknown and difficult to derive. Therefore, in FAKF, current monitoring data are introduced to calculate $V_{k}^{-}$ through the current–voltage model that can be derived from the Direct Current (DC) motor equation. The DC motor equation can be represented as follows [48]:(13)$$\nu_{m}=K_{e}\omega_{m}+i_{m}R+L\frac{di_{m}}{dt},$$
(14)$$K_{t}i_{m}-T_{ex}=B\omega_{m}+J\frac{d\omega_{m}}{dt},$$
where $\nu_{m}$, $i_{m}$, $\omega_{m}$, *L* and $T_{ex}$ denote armature voltage, armature current, armature angular speed, armature inductance and external torque load, respectively. $J=J_{m}+J_{L}$ represents the net system inertia and $B=B_{m}+B_{L}$ represents the net viscous friction coefficient (with contributions from the motor and the load). $K_{e}$ and $K_{t}$ denote the back-emf constant and the electromagnetic torque constant, respectively. Based on Equations (Equation 13) and (Equation 14), the current–voltage model in Z domain with the order of 2 can be obtained as follows:(15)$$\frac{V_{m}\left(z\right)}{I_{m}\left(z\right)}=\frac{b_{0}+b_{1}z^{-1}+b_{2}z^{-2}}{1+a_{0}z^{-1}+a_{1}z^{-2}},$$
where $a_{0}$, $a_{1}$, $b_{0}$, $b_{1}$, $b_{2}$ are the model parameters determined by EMA motor voltage and current monitoring data as well as a model identification algorithm. In this way, the KF transition equation for EMA motor voltage is substituted by this current–voltage model. Through replacing $V_{k}^{-}$ with $IM_{k}$, the voltage estimation at current time step *k* can be denoted as follows:(16)$$V_{k}=g\left(IM_{k},VM_{k}\right)=f\left\{Z^{-1}\left[I_{m}\left(z\right)\left(\frac{b_{0}+b_{1}z^{-1}+b_{2}z^{-2}}{1+a_{0}z^{-1}+a_{1}z^{-2}}\right)\right],VM_{k}\right\},$$
where $V_{k}$ and $VM_{k}$ are the same as Equation (Equation 12). $IM_{k}$ and $I_{m}$(z) represent the current measurement at current time step *k* and its Z-transform of current measurement. g(•) indicates the function of $V_{k}$ and $IM_{k}$, $VM_{k}$. $Z^{-1}\left(•\right)$ denotes the Inverse Transform in Z domain. Based on Equation (Equation 16), it can be known that the $V_{k}^{-}$ is substituted by the $IM_{k}$ and the voltage estimation $V_{k}$ can be directly obtained through $IM_{k}$, $VM_{k}$ rather than through $V_{k}^{-}$, $VM_{k}$. As a result, the KF state equation for EMA voltage estimation can be derived.

## 3. The Framework of FAKF-Based Voltage Estimation

In this study, a new framework named FAKF-based voltage estimation is proposed to implement EMA motor voltage estimation, which is shown in Figure 1.

In this framework, a physical model of current and voltage is introduced to substitute a KF state equation, which is required in KF-based EMA voltage estimation but not available. In FAKF-based voltage estimation, firstly, the monitoring data of EMA motor current and voltage should be acquired. Secondly, a physical model of current and voltage requires to be constructed based on a basic principle of EMA and its parameters through utilizing monitoring data and model parameter identification algorithms. In this study, a classical model identification algorithm, Least Squares Criterion identification algorithm, is used to estimate model parameters, which can meet the accuracy requirements of voltage estimation. There are also some good model identification algorithms which can be considered when applying an FAKF-based estimation method in others applications with higher requirements of accuracy [49,50]. Finally, applying the KF method to achieve high-accuracy voltage estimation of EMA motor, the built physical model is used to substitute the KF state equation. Specifically, five steps are included in the framework of FAKF-based EMA motor voltage estimation.

**Step 1 (Data Acquisition):** the monitoring data of feature to be estimated and the related feature is required to be obtained. In this study, voltage is the feature to be estimated while current is chosen to be the related feature.

**Step 2 (Model construction):** the model structure of the feature to be estimated and the related feature needs to be determined. Generally, the transfer function of the two features can be easily obtained. The model structure of voltage and current utilized in this study is presented in Equation (Equation 15).

**Step 3 (Model Identification):** model parameters are derived from the monitoring data based on model identification algorithm. The Least Squares Criterion identification algorithm is adopted to identify the model parameters.

**Step 4 (KF-based Estimation):** the model is determined after model construction and model identification, which can be utilized as a KF transition equation. Based on voltage monitoring data and estimation data with the constructed model, KF-based voltage estimation can be implemented.

**Step 5 (Estimation data Output):** the estimated feature is the output of FAKF. In this study, the estimated voltage is set to be the output of FAKF-based voltage estimation which can be further utilized in EMA PHM.

## 4. Experiment Results and Analysis

### 4.1. FLEA Introduction

To validate the effectiveness of the proposed method, the published data derived from Flyable Electro-Mechanical Actuator (FLEA) test stand are utilized in this study. NASA Prognostics Center of Excellence (PCoE) make the FLEA test stand. Three distinct actuators are included in FLEA (i.e., the normal actuator, the faulty actuator and the load actuator). The load actuator is utilized to provide a dynamic load for the faulty actuator or the normal actuator. Through changing the load from the normal one to the faulty one, EMA fault injection experiments can be implemented without replacing actuators. FLEA can also be connected to aircraft via data bus. By this way, the motion and load profile from aircraft can be directly utilized to drive FLEA. The model of FLEA is presented in Figure 2a. A multi-axis motion controller is utilized to control the three actuators mentioned above. The electro-magnets utilized in FLEA between the load and two test actuators guarantee that the load actuator is only coupled to one test actuator. Figure 2b shows the UltraMotion Bug Actuator utilized in FLEA as the test actuator and load actuator. These stand components of FLEA enable EMA run-to-failure experiments possible and affordable. Figure 2c shows the actual FLEA.

### 4.2. Data Description of EMA

The FLEA data acquisition system is mainly composed of two NI 6259 measuring modules and many sensors. There are two channels in the data acquisition system with different sample rates. The lower one (1 kHz) is utilized to obtain the data of voltage, current, temperature, etc., while the higher one (20 kHz) is for vibration monitoring. The low-speed data of FLEA actuator X (faulty actuator) with jams fault under negative load and positive load are shown in Figure 3 and Figure 4, respectively.

Obviously, six parameters (i.e., Measured Load, Motor X current, Motor X Voltage, Motor X Temperature, Nut X Temperature, Ambient Temperature) are directly related to the faulty actuator—X actuator.

### 4.3. Voltage Estimation Based on FAKF

EMA motor voltage estimation is critical for EMA PHM while the physical model about voltage and time required for the model-based estimation method cannot be obtained. Simultaneously, the data-driven estimation method is not accurate enough due to the existing EMIs in the EMA motor. Thus, KF which has a good performance on parameter estimation is focused for EMA motor voltage estimation in this work. However, the state equation of EMA voltage required for KF-based estimation method is not available due to the complexity of EMA. Thus, the KF cannot be directly utilized in EMA voltage estimation. To meet the requirement for accurate EMA voltage estimation, a new method named FAKF is proposed in which the physical model about EMA current and voltage is utilized to substitute the state equation in KF. In this study, model-based voltage estimation means that the physical model of current and voltage is directly utilized in the process of voltage estimation. To prove the effectiveness of the FAKF-based estimation method for EMA motor voltage, experiments with FAKF-based voltage estimation and model-based voltage estimation is conducted, in which NASA FLEA published data are utilized. The flowchart of FAKF-based EMA motor voltage estimation is presented in Figure 5.

Firstly, the FLEA data are collected, which are derived from *N* work cycles. Then, a suitable model structure is constructed and its model parameters are identified utilizing the data of the first work cycle. Next, the identified model is utilized as the KF transition equation. Finally, based on the proposed FAKF method and related monitoring data, accurate voltage estimations are obtained.

As mentioned above, NASA FLEA data are utilized in this experiment. The voltage and current data of the first work cycle under negative load and positive load are shown in Figure 6, Figure 7, Figure 8 and Figure 9, respectively.

The parameters are identified based on voltage and current data of the first work cycle. The identified results are presented in Table 1.

To make the voltage estimation curve of model-based method and FAKF-based method clearer, the results are shown on a short time axis (i.e., 0.15 s). The output of model-based voltage estimation under different loads are presented in Figure 10 and Figure 11. Obvious deviation between model-based estimation and voltage monitoring data can be seen.

Figure 12 and Figure 13 show the estimation results utilizing the FAKF-based method and model-based method. The curve of FAKF-based voltage estimation is very close to voltage monitoring data.

The model parameters utilized in this study are derived from the data of the first work cycle, and they may not match the data of other work cycles as time-varying EMI exists. Therefore, the model-based voltage estimation is of worse quality than FAKF-based voltage estimation. When treating voltage monitoring data as the substitute of real voltage value, the performance of these two methods can be evaluated utilizing mean absolute error (MAE) and root mean squared error (RMSE). MAE and RMSE of estimation results are presented in Table 2 and Table 3.

In contrast to model-based estimation, the MAE of FAKF-based estimation with −40 lbs and +40 load are decreased by 55.2% and 56.8%, respectively. RMSE of FAKF-based estimation with −40 lbs and +40 load are decreased by 64.5% and 60.8%. It can also be seen from Table 2 and Table 3 that MAE and RMSE of FAKF-based estimation under negative and positive load are both smaller than model-based estimation. In fact, the introduced current monitoring data make the voltage estimation of FAKF-based method closer to the real voltage value, which is the key reason of the reduction of MAE and RMSE for FAKF-based voltage estimation. As for the time consumption of FAKF-based estimation method, it can mainly be divided into two stages (i.e., model training and model application ). In this study, the operating time of them is obtained through running this method on a PC. For model training, its operating time is 0.013 s with 2936 points. In addition, for model application, its operating time is 9.985 s with 92,415 points. The operating time for each sample point in the model application is about 0.0001 s, which is far less than the sampling period of NASA FLEA data. Thus, in this study, the FAKF-based estimation method can be considered to be a real-time method for FLEA motor voltage estimation. Therefore, it can be concluded that the performance of the FAKF-based method in voltage estimation is better than the model-based method. Accurate voltage estimation obtained by utilizing an FAKF-based method will contribute a lot to EMA voltage-based PHM studies, such as Health Indicator (HI) extraction, degradation modelling and remaining useful life prediction.

## 5. Conclusions

This study proposes a new method named FAKF for EMA motor voltage estimation. In FAKF, a physical model is introduced as the transition equation in KF to enhance the accuracy of estimation. To validate the effectiveness of FAKF-based voltage estimation, experiments based on the FAKF method are conducted, in which the monitoring data with negative and positive load from NASA FLEA are utilized. Experiment results show that FAKF has a better performance for EMA motor voltage estimation than the model-based method. Under −40 lbs load condition, MAE and RMSE of FAKF-based voltage estimation are decreased by 55.2% and 64.5%, while, under +40 lbs load condition, MAE and RMSE of FAKF-based voltage estimation are decreased by 56.8% and 60.8%. The proposed FAKF method can be utilized to estimate EMA voltage under positive and negative load and it also provides an effective solution for parameter estimation of other complex systems. In addition, further studies of voltage-based PHM, such as HI extraction, degradation modelling and remaining useful life prediction, will be more reliable with the FAKF-based voltage estimation.

However, the practical application of the method in PHM is less considered, such as degradation modelling, remaining useful life prediction and applying the FAKF-based estimation method into embedded platforms. Complex physical models about the EMA motor parameters and high-quality model parameter identification algorithms can also be further studied to improve the performance of FAKF. Moreover, non-Gaussian noise has not been fully considered in the proposed method. In our future studies, the research about these challenges will be focused [51,52].

## Figures and Tables

**Figure 1 sensors-18-04190-f001:**
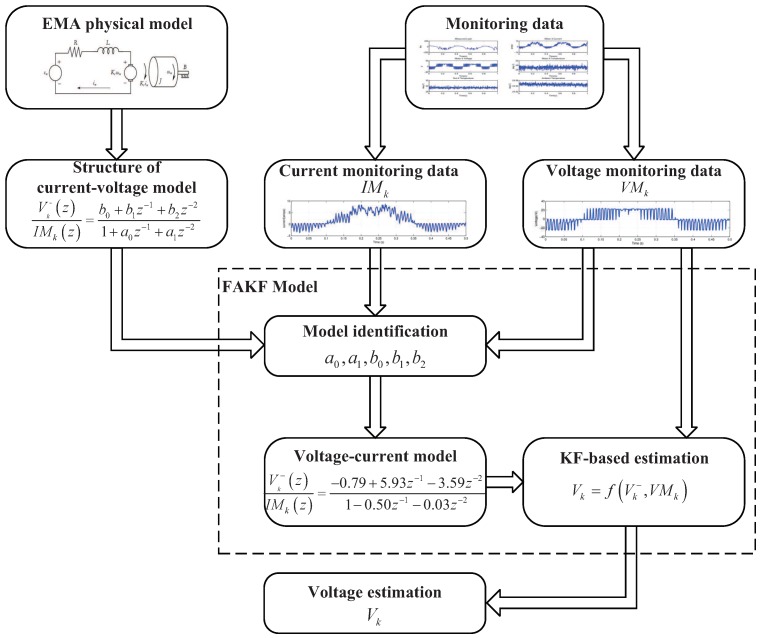
FAKF-based EMA motor voltage estimation.

**Figure 2 sensors-18-04190-f002:**
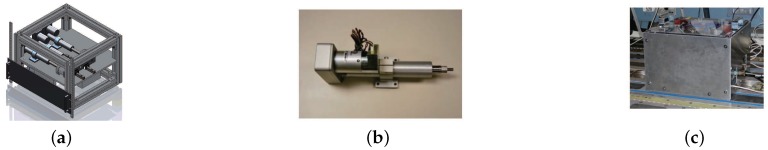
(**a**) FLEA model; (**b**) UltraMotion bug actuator; (**c**) actual FLEA [1].

**Figure 3 sensors-18-04190-f003:**
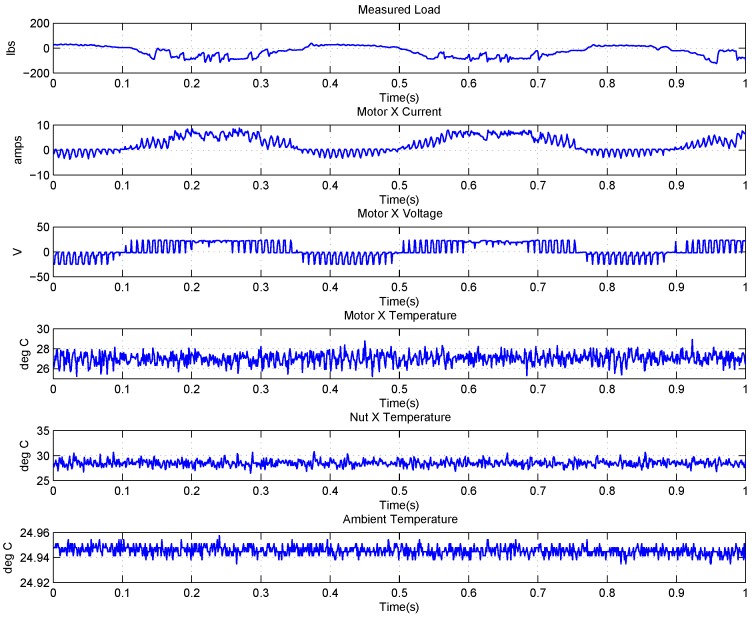
Data overview of FLEA X actuator with −40 lbs load.

**Figure 4 sensors-18-04190-f004:**
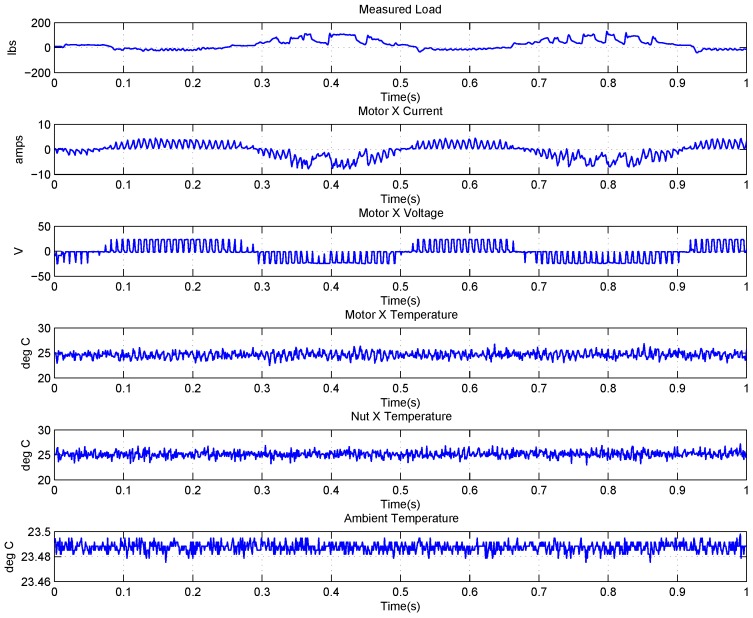
Data overview of FLEA X actuator with +40 lbs load.

**Figure 5 sensors-18-04190-f005:**
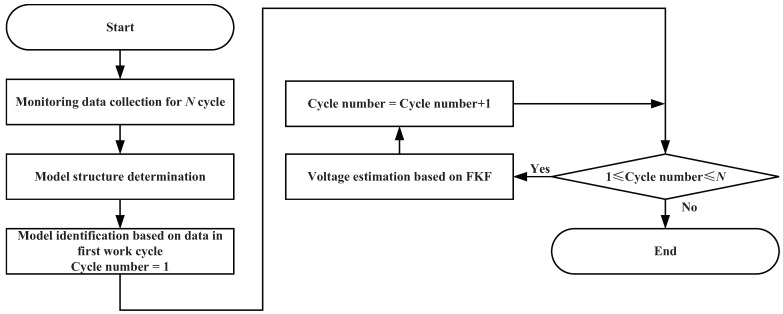
FAKF-based voltage estimation flowchart.

**Figure 6 sensors-18-04190-f006:**
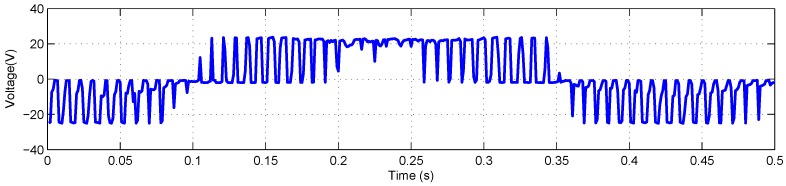
Voltage monitoring data of the first work cycle with −40 lbs load.

**Figure 7 sensors-18-04190-f007:**
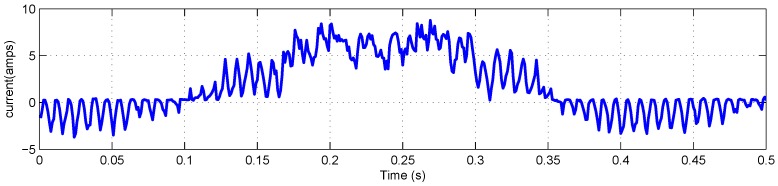
Current monitoring data of the first work cycle with −40 lbs load.

**Figure 8 sensors-18-04190-f008:**
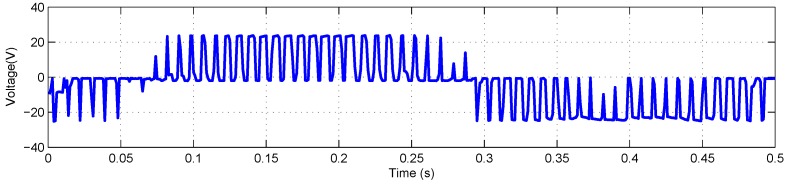
Voltage monitoring data of the first work cycle with +40 lbs load.

**Figure 9 sensors-18-04190-f009:**
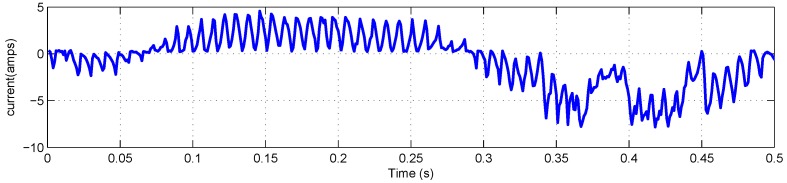
Current monitoring data of the first work cycle with +40 lbs load.

**Figure 10 sensors-18-04190-f010:**
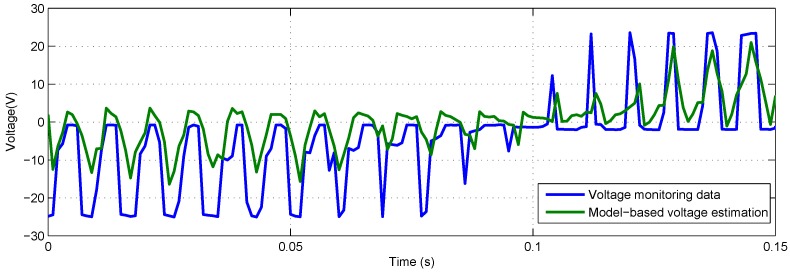
Model-based voltage estimation and voltage monitoring data with −40 lbs load.

**Figure 11 sensors-18-04190-f011:**
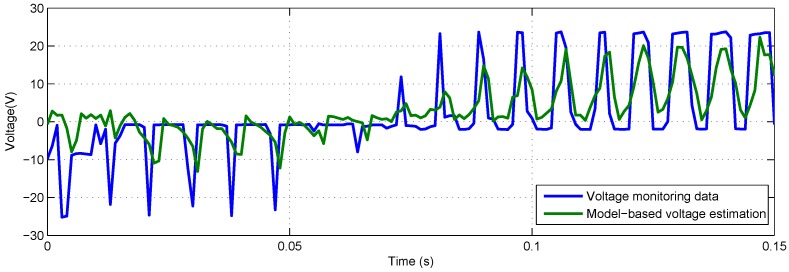
Model-based voltage estimation and voltage monitoring data with +40 lbs load.

**Figure 12 sensors-18-04190-f012:**
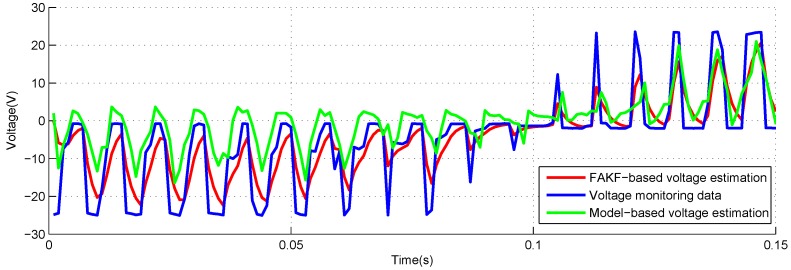
Voltage estimation based on FAKF with −40 lbs load.

**Figure 13 sensors-18-04190-f013:**
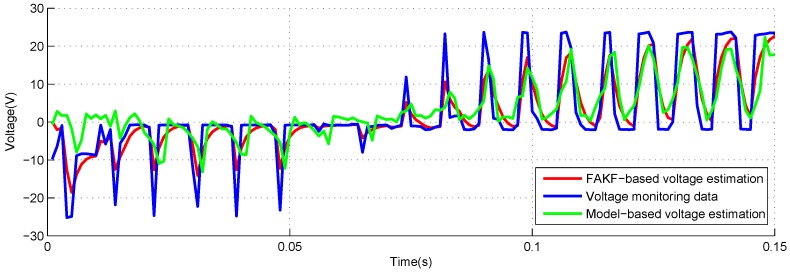
Voltage estimation based on FAKF with +40 lbs load.

**Table 1 sensors-18-04190-t001:** Parameter identification results.

Parameters	Identified Results with −40 lbs Load	Identified Results with +40 lbs Load
$a_{0}$	−0.4998	−0.4424
$a_{1}$	−0.0343	−0.0512
$b_{0}$	−0.7889	−1.2471
$b_{1}$	5.9267	6.7634
$b_{2}$	−3.5857	−3.2269

**Table 2 sensors-18-04190-t002:** MAE and RMSE of FAKF-based estimation and model-based estimation with −40 lbs load.

	FAKF-Based Estimation	Model-Based Estimation
**MAE**	3.4361	6.2292
**RMSE**	5.1754	8.0186

**Table 3 sensors-18-04190-t003:** MAE and RMSE of FAKF-based estimation and model-based estimation with +40 lbs load.

	FAKF-Based Estimation	Model-Based Estimation
**MAE**	3.7047	6.5225
**RMSE**	5.1613	8.4893

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
