# Peer review of "An Electro-Mechanical Actuator Motor Voltage Estimation Method with a Feature-Aided Kalman Filter†"

_sensors, 2018, doi:10.3390/s18124190_

Reviewer 1 Report

The paper deals with a method based on FAKF aiming to reduce deviation exists between motor voltage monitoring data and real motor voltage due to electromagnetic interference. The authors present the proposed model followed by experimental tests in order to evaluate de method. Generally speaking, the paper is in good quality. I have just a few comments to the authors:

1- I suggest a deeper discussion about the differences found between the real model and FAKF. The authors have only presented the results without presenting any discussion about the drawbacks of the model.

2- In general modelling is a time-consuming task. Could you provide information about the time consumption for your method? Could the method work in real time?

Author Response

Please find the point-by-point responses in the attachment.

Reviewer 2 Report

The paper is very good in which a Feature-aided Kalman Filter (FAKF) method is proposed, in which the state equation is substituted by a physical model of current and voltage.
To improve the paper please add some fundamental aspects on FAKF method to help the Readers to understand even though they do not know this theoretical aspects.
In this context equation (13)  should be clarified in the paper. In particular, the order of the Transfer function must be discusse.
Equation (14) should be discussed. In fact, it seems to be too abstract.
Observing the results, a question arrises: How did you tun the Kalman Filter?
Please explain this very important aspect.
How did you estimate the Parameter of the model?
Concerning the cited literature, it could be useful to consider the following paper which wer related to the same topic.
Concerning the parameter of the model, in the following papers the parameter of a PMSM is considered in which a method to estimate the Parameters of a synchronous electrical Maschine with permanent magnets is considered. This method can be used to "help" a Kalman Filter.
Mercorelli, P. Parameters identification in a permanent magnet three-phase synchronous motor of a city-bus for an intelligent drive assistant (2014) International Journal of Modelling, Identification and Control, 21 (4), pp. 352-361.Mercorelli, P. A decoupling dynamic estimator for online parameters indentification of permanent magnet three-phase synchronous motors (2012) IFAC Proceedings Volumes (IFAC-PapersOnline), 16 (PART 1), pp. 757-762.Other kinds of observers for electromagnetic actuators and Motors  can be found also here which can be useful in the technical context of the introduction of the paper. Hilairet, M., Auger, F., Berthelot, E. Speed and rotor flux estimation of induction machines using a two-stage extended Kalman filter (2009) Automatica, 45 (8), pp. 1819-1827. Mercorelli, P. A Motion-Sensorless Control for Intake Valves in Combustion Engines (2017) IEEE Transactions on Industrial Electronics, 64 (4), art. no. 7534788, pp. 3402-3412.Vieira, R.P., Gastaldini, C.C., Azzolin, R.Z., Gründling, H.A. Sensorless sliding-mode rotor speed observer of induction machines based on magnetizing current estimation (2014) IEEE Transactions on Industrial Electronics, 61 (9), art. no. 6671412, pp. 4573-4582.
In the context of external estimation of unaccessible disturbances you can consider the paper of chen and the literature therein:
Chen, L., Mercorelli, P., Liu, S. A Kalman estimator for detecting repetitive disturbances (2005) Proceedings of the American Control Conference, 3, pp. 1631-1636.

Author Response

Please find the point-by-point responses in the attachment.
